# Effects of presurgical interventions on chronic pain after total knee replacement: a systematic review and meta-analysis of randomised controlled trials

Jane Dennis  ,[1] Vikki Wylde,[1,2] Rachael Gooberman-Hill,[1,2] AW Blom,[1,2] Andrew David Beswick  [1]

¹Musculoskeletal Research Unit, Translational Health Sciences, University of Bristol, Bristol, UK
²National Institute for Health Research Bristol Biomedical Research Centre, University Hospitals Bristol NHS Foundation Trust and University of Bristol, Bristol, UK

**Correspondence to**
Dr Jane Dennis;
jane.dennis@bristol.ac.uk

## ABSTRACT

**Objective** Nearly 100 000 primary total knee replacements (TKR) are performed in the UK annually. The primary aim of TKR is pain relief, but 10%–34% of patients report chronic pain. The aim of this systematic review was to evaluate the effectiveness of presurgical interventions in preventing chronic pain after TKR.

**Design** MEDLINE, Embase, CINAHL, *The Cochrane Library* and PsycINFO were searched from inception to December 2018. Screening and data extraction were performed by two authors. Meta-analysis was conducted using a random effects model. Risk of bias was assessed using the Cochrane tool and quality of evidence was assessed by Grading of Recommendations Assessment, Development and Evaluation.

**Primary and secondary outcomes** Pain at 6 months or longer; adverse events.

**Interventions** Presurgical interventions aimed at improving TKR outcomes.

**Results** Eight randomised controlled trials (RCTs) with data from 960 participants were included. The studies involved nine eligible comparisons. We found moderate-quality evidence of no effect of exercise programmes on chronic pain after TKR, based on a meta-analysis of 6 interventions with 229 participants (standardised mean difference 0.20, 95% CI −0.06 to 0.47, I²=0%). Sensitivity analysis restricted to studies at overall low risk of bias confirmed findings. Another RCT of exercise with no data available for meta-analysis showed no benefit. Studies evaluating combined exercise and education intervention (n=1) and education alone (n=1) suggested similar findings. Adverse event data were reported by most studies, but events were too few to draw conclusions.

**Conclusions** We found low to moderate-quality evidence to suggest that neither preoperative exercise, education nor a combination of both is effective in preventing chronic pain after TKR. This review also identified a lack of evaluations of other preoperative interventions, such as multimodal pain management, which may improve long-term pain outcomes after TKR.

**PROSPERO registration number** CRD42017041382.

### Strengths and limitations of this study

► This systematic review evaluates recent evidence of the effectiveness of presurgical interventions in preventing chronic pain after total knee replacement.
► Synthesis of adverse events data was not possible.
► We only included studies that completed intervention delivery during the preoperative period and did not include studies that evaluated interventions that began in the preoperative period but extended into the perioperative and postoperative period.

## BACKGROUND

Osteoarthritis is the most common condition that affects the knee joint and causes considerable pain and disability. Effective treatments noted in guidelines include pain medications, exercise and weight loss when appropriate.[1 2] If symptoms do not respond to pharmacological and conservative treatments, people may receive a total knee replacement (TKR). Annually in the UK, nearly 100 000 primary TKRs are performed,[3 4] and in the USA in 2010, about five million people were estimated to be living with a TKR.[5] Outcomes are good for many, but a systematic review found that 10%–34% of patients report unfavourable long-term pain outcomes at between 3 months and 5 years after TKR[6] which is associated with dissatisfaction with the outcome of surgery.[7 8]

Some patient factors are associated with poor long-term pain outcomes. Using structural equation modelling, Sayers and colleagues demonstrated that pain during the presurgical period, particularly on movement, is strongly associated with chronic postsurgical pain, and a potentially important target for intervention.[9] Other presurgical risk factors for pain identified in multivariable analyses in representative populations include: poorer physical

function, body mass index ≥35 kg/m$^2$ and poor general mental health.[10] With knowledge of risk factors for chronic pain, there is the potential for targeting care to those at high risk or for risk factor modification.[10 11] While many presurgical interventions focus on preparing the patient for the procedure and hospital stay, reducing perioperative pain and facilitating early mobilisation and recovery,[10] their impact on the key outcome of chronic pain remains to be established. This is achievable through high quality, adequately powered randomised controlled trials with long-term follow-up for patient-related pain outcomes and ultimately through well-conducted systematic reviews with thorough consideration of sources of bias that may influence study results.

In a systematic literature search of MEDLINE, Embase, PsycINFO, CINAHL and *The Cochrane Library* on 18 December 2018, we identified 36 systematic reviews of preoperative interventions in TKR. Twenty-five of these had searches conducted more than 5 years previously,[10 12–35] and nine exclusively reported short-term outcomes (3 months or less after surgery).[36–44] Two more recent reviews considered the outcome of long-term pain[45 46] but neither had a registered protocol. Chesham and Shanmugam identified two randomised controlled trials in their search window (2004–2014) that reported no difference in pain at 6 months after surgery in patients randomised to exercise or control.[45] Wang and colleagues identified three randomised controlled trials of preoperative exercise-based interventions in patients waiting for knee or hip replacement that reported long-term pain and were published up to November 2015.[46] No difference in pain outcome was apparent at 6 months or longer in patients receiving intervention or control.

In this systematic review and meta-analysis, we assessed the effectiveness of preoperative interventions in preventing chronic pain in patients receiving TKR. Previous reviews have largely focused on specific interventions but we have opted for a temporal framework and sought to include any intervention conducted before surgery. This review is part of a broad suite of reviews considering the timing of intervention throughout the TKR care trajectory.[47 48]

## METHODS

The protocol for this systematic review was registered with PROSPERO.[49] We formulated the research question according to the participants, intervention(s), comparator(s), outcomes (PICO) principle[50] and used methods based on those recommended by Cochrane.[51] Reporting has been in accordance with the Preferred Reporting Items for Systematic Reviews and Meta-Analyses (PRISMA) guidelines,[52] and a checklist is included in the online supplementary appendix.

### Patient and public involvement

Before starting the review, we held a meeting with stakeholders and patient representatives and discussed inclusion criteria and outcomes. Areas of potential relevance for presurgical intervention were highlighted, specifically exercise, education, psychological therapies, weight management, nutritional supplements, management of comorbidities and pain management with intra-articular injections. The importance of patient-reported pain as an outcome after TKR was emphasised. A musculoskeletal patient and public involvement forum has discussed the review and will advise on plain language summaries and dissemination.[53]

### Types of studies

To limit selection bias, we included only randomised controlled trials. Studies reported only as abstracts or unobtainable as full text copies were excluded. Language was not an exclusion criterion.

### Participants

Eligible participants were adults on the waiting list for TKR, up until the point of admission for surgery.

### Interventions

Preventive interventions (pharmacological or non-pharmacological) delivered during the presurgical period were included. Studies where delivery of the intervention extended into the postoperative period were excluded.

### Comparator group

Comparator group participants included those who had received usual care or an alternative intervention.

### Outcomes

In preference, we considered patient-reported joint-specific pain intensity, typically measured using pain domains of the Oxford Knee Score,[54] Western Ontario and McMaster Universities Osteoarthritis Index (WOMAC)[55] or Knee injury and Osteoarthritis Outcome Score (KOOS)[56] at 6 months or longer after knee replacement. We selected these outcomes in preference to pain components of Knee Society's Clinical and Functional Scoring System (KSS)[57] and Hospital for Special Surgery Knee Score (HSS)[58] in which pain assessment is limited to one or two questions on intensity[59] and relies on clinician assessment which may not reflect the views of patients.[60] The choice of outcome timing reflects the time when, for many, pain levels have optimised.[61] If joint specific measures were not available, then we planned to use pain dimensions from quality of life measures including Short Form 36 Health Survey (SF-36)[62] and Short Form 12 Health Survey.[63] Had neither joint-specific nor generic pain outcomes been available, we planned to use data for pain measured using a Visual Analogue Scale (VAS) or Numerical Rating Scale. When no patient-reported outcome measure was reported, we used pain data from surgeon-assessed scores. When discrete pain data were unavailable, we reported overall data that included those relating to pain, typically in combination with function and other factors.

Where data were available, we planned to report proportions of patients with a favourable pain outcome (eg, 'no worse than mild pain') as recommended in clinical trials[64] and systematic reviews.[65]

Data on all adverse events were extracted and summarised narratively. Our focus was on adverse events related to the intervention, although we also report those related to surgery.

## Study design

Randomised controlled trials of interventions commencing and completing within the presurgical period, with follow-up at 6 months or longer after surgery for TKR, were identified and assessed for eligibility. Authors reporting long-term outcomes of any type (typically, function only) were contacted to check whether pain outcome data were also available; those reporting aggregated data for hip replacements together with knee replacements were also contacted. Studies meeting these criteria and reporting pain data or composite scores including pain were included.

## Database searches

We established a database of all randomised controlled trials in knee replacement, in preparation for a suite of systematic reviews in chronic pain after TKR.[47 48] Relevant trials were identified through searches (last updated on 18 December 2018) in: MEDLINE, Embase and PsycINFO on OvidSP; CINAHL on EBSCOhost; and *The Cochrane Library*. The search strategy as applied in MEDLINE is shown in online supplementary appendix. Within this resource, we searched for interventions conducted in the presurgical setting.

Citations and reference lists of key reviews and randomised trials were checked in ISI Web of Science. The ISRTN registry on BioMed Central was searched for ongoing randomised trials which were then checked for subsequent publication. No language restrictions were applied, and data from potentially relevant non-English language articles were extracted and translated to establish eligibility. Studies reported only as abstracts or that we were unable to acquire as full text copies using interlibrary loans or author contact were excluded.

## Study screening and data extraction

Records identified by searches were imported into Endnote V.X9 (Thomson Reuters). An initial screen for potential eligibility was undertaken by one review author (JD) with articles excluded only if clearly irrelevant. Typically, these were studies in osteoarthritis with no surgery, arthroscopy only or surgery involving joints other than the knee. Subsequently, abstracts and full text articles were screened independently by two review authors. After consensus, reasons for exclusion were recorded.

Data from relevant randomised trials were extracted by one review author onto forms which were tested and refined before use. Microsoft Excel was used to record abbreviated information, specifically on: country where study was conducted; dates of recruitment; participant characteristics (indication, age, sex); inclusion and exclusion criteria; intervention and control content; setting, timing, duration and intensity of intervention; follow-up data collection (including loss to follow-up); instruments used to collect data on pain; outcome data including pain and adverse events. Data extracted were checked against source material by a second review author.

Group means and SDs of pain outcome measures were recorded at the longest follow-up interval. Data with which to calculate proportions of patients with a poor outcome, however defined, were sought. Authors were contacted for unpublished data. Details of adverse events of any type were extracted.

## Risk of bias within included studies

Potential sources of bias were assessed using the Cochrane risk of bias tool[66] by two review authors (JD and ADB) working independently; disagreements were adjudicated by a third (VW). We present analyses where possible showing data from all relevant studies for a given outcome, but also conducted sensitivity analyses excluding studies assessed to be at overall high risk of bias.

## Data analysis

When studies reported pain outcome data, these were presented and pooled where possible, using Review Manager V.5.0.[67] As the control group in one study was shared by two active interventions, the sample size in meta-analysis was halved for the control group, to preserve independence of findings.[51] Results are presented for one comparison in the form of standardised mean differences (SMDs), as instruments used to measure outcomes varied.

A random effects model was planned, due to anticipated heterogeneity as described below. A sensitivity analysis excluding those comparisons for which we had imputed data and/or studies we had assessed as being at overall high risk of bias was conducted. At protocol stage,[49] we planned to generate funnel plots, if ten or more studies with similar interventions and controls were identified, as a method of estimating the risk of publication bias. In the event, data were not sufficient to do so.

Heterogeneity between studies (considering both magnitude and direction of effect) was assessed using visual inspection of the graph and the I² statistic. We anticipated sources of heterogeneity justifying subgroup analyses might include format and intensity of intervention; however, statistical heterogeneity proved to be unimportant, and data were insufficient to conduct such analyses.

## Grading of Recommendations Assessment, Development and Evaluation approach to evidence

We used the Grading of Recommendations Assessment, Development and Evaluation (GRADE) approach to assess the quality of the evidence,[68] specifically study

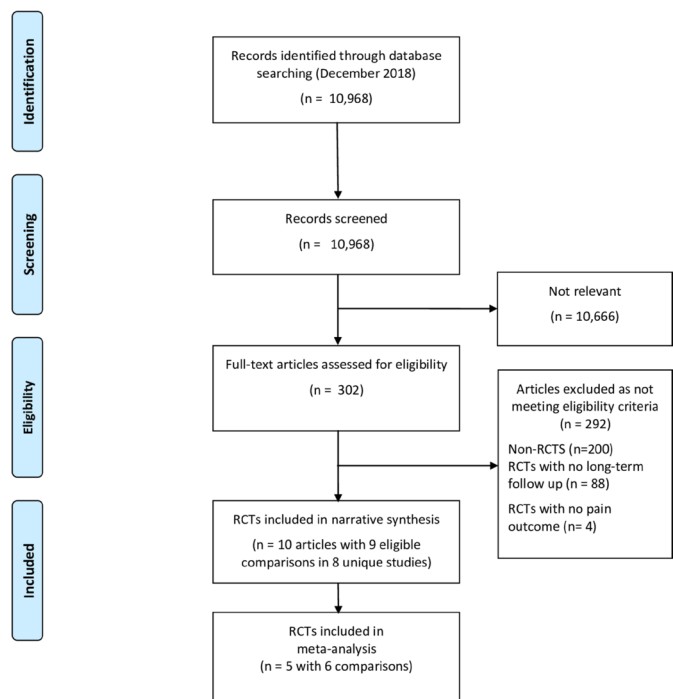

**Figure 1** Preferred Reporting Items for Systematic Reviews and Meta-Analyses flow diagram.

limitations, consistency of effect, imprecision, indirectness and publication bias.

## RESULTS

Conduct of the review is summarised in figure 1. Searches identified 10 968 articles. Of these, 302 potentially relevant reports were considered in detail, of which 100 were identified as RCTs (some with multiple publications). Eighty-eight did not follow up participants to a minimum of 6 months. Four trials involving different interventions (smoking cessation,[69] testosterone injections,[70] epoetin-α injections[71] and bacterial screening[72] followed by decolonisation) did meet eligibility criteria for long-term follow-up, but none of these included outcome data related to pain.

After detailed evaluation, 10 articles reporting data from 8 RCTs[73–80] were included in the review. These 8 RCTs included data from 960 participants randomised to nine eligible comparisons. The interventions evaluated in the included studies were exercise, education or a combination of both.

Seven publications assessed the impact of a single intervention (exercise (5), education (1) or a combination of both exercise and education (1)) against a no-treatment or attentional control. One publication reported relative effects of two different physical interventions (cardio-based exercise or physiotherapy) against a no treatment control.[73] We analysed data from the latter publication as two separate studies, dividing the control group during analysis so as to preserve independence of findings. Details of studies, including intervention characteristics, are shown in table 1; risk of bias assessments in figure 2;

meta-analysis and sensitivity analyses in figures 3–6. Finally, GRADE assessments of the quality of evidence for these classes of intervention for the outcomes of pain and serious adverse events are shown in table 2.

### Overview of study settings, sample sizes and interventions

Included studies were published between 1996 and 2018. Two studies were conducted in Canada,[74 80] two in the USA,[73 79] and one each in Belgium,[77] Denmark,[75] Switzerland[76] and Thailand.[78] Exercise interventions were delivered within clinics or within the home; education interventions were delivered within clinics in the studies in which it formed part of a composite intervention, or online when it was assessed in isolation.

Samples at recruitment ranged from 30 to 416 participants. The total number of participants involved in studies of exercise alone was 373; exercise and education together, 131; and education alone, 416.

Sample size calculations were reported by most investigators—but only one was based on the primary outcome used within this review (pain at 6 months). Other studies had as their main focus: satisfaction with TKR[80]; a short-term reduction of 10 points on the HSS and the reduction of hospital stay by 1 day[73]; the Chair Stand test[76]; a difference of 10° in passive flexion[77]; an 8-point difference in WOMAC[79]; and a 10-point difference on the KOOS ADL subscale.[75] The variability in foci among studies makes it difficult for us to estimate a reasonable optimal information size for the review, particularly as some investigators revised their sample size estimates upwards as part of considering limitations of their studies.[73 74]

### Participants

Five studies exclusively recruited participants undergoing TKR for osteoarthritis[74 76–78 80]; one permitted those with rheumatoid arthritis[73]; and two included participants undergoing either hip or knee replacement for osteoarthritis, with long-term data reported separately[79] or in aggregate.[75] Mean age of participants ranged from 63 to 70 across studies. Gender, where reported, varied greatly, with studies recruiting 17%[78] to 60% women.[73]

### Interventions

Seven interventions assessed within six studies[73 75–79] were exercise-based; one was multifactorial with exercise and an education component focusing on walking with crutches, bed mobility and postoperative range of movement[74]; and one featured an online educational component alone.[80]

All exercise interventions included strengthening,[73–79] with additional components targeting gait re-education,[74] functional exercise,[74 76] improving knee range of movement, flexibility or mobility[73 74 76 77 79] and cardiovascular conditioning.[73 76 79] Two studies evaluated a particular manualised programme, 'NEuroMuscular Exercise training program for patients with knee or hip osteoarthritis assigned for total joint replacement' (NEMEX-TJR)[81] which is described as a series of

**Table 1** Characteristics of studies evaluating preoperative exercise and education

| Publication Location Date of study | Indication Number randomised (intervention: control) Mean age (%female) | Primary focus of intervention | Study setting Timing, duration and intensity | Control group care | Longest follow-up after surgery Losses to follow-up (intervention: control) risk of bias | Outcomes |
|---|---|---|---|---|---|---|
| Beaupre et al,[74] 2004 Canada Date not specified | Non-inflammatory arthritis n=131 (65:66) 67 years (55%) | Multifactorial: exercise (gait re-education, functional, ROM, strengthening) and education on crutch walking, bed mobility and postoperative ROM. | Community physical therapy clinic group 3-times per week for 4 weeks, 6 weeks before surgery. | No intervention | 6 and 12 months 16 (10:6) patients cancelled surgery, 6 (4:2) patients lost to 12 month follow-up. Low overall risk of bias. | No difference in WOMAC pain or SF-36 bodily pain at 6 and 12 months (p>0.05). General complications, 'similarly dispersed across both groups': Pulmonary embolism (2:2), DVT (3:6), superficial infection (2:3), deep infection (1:0), manipulation (1:2). |
| Culliton et al,[80] 2018 Canada 2013–2014 | Osteoarthritis n=416 (207:209) 64 years (43%) | Education: access to an online e-learning tool during preadmission visit and 31-page guide also provided to the control group. Tool consisted of custom-made 32 videos (1–2 min) addressing patient expectations, presented by surgeons, physical therapists and patients. | Access to e-learning tool remained active until the study end (1 year). Tool consisted of custom-made 32 videos (1–2 mins) addressing patient expectations, presented by surgeons, physical therapists and patients. | Booklet alone | 12 months postoperative. Data analysed for 76.6% of original sample randomised. Reasons include TKR not taking place or lack of email address. Later exclusions due to failure to submit data. Low overall risk of bias. | 1 year postoperative, significant between-group differences in favour of the control group for the KOOS symptoms score (p=0.04). Pain scores obtained from authors directly were not significantly different between groups. Adverse events not reported. |
| D'Lima et al,[73] 1996 Cardiovascular conditioning USA Date not specified | Rheumatoid arthritis or osteoarthritis n=20 (10:10) 70.6 years (35%) | Exercise (stretching, strengthening, cardiovascular conditioning). | Hospital. Individualised 45 min sessions, 3 times a week for 18 weeks commencing 6 weeks before surgery. | One meeting with physical therapist | 24 and 48 weeks. No losses to follow-up reported. High risk of bias: small numbers of patients; baseline demographic differences; failure to report data other than graphically. | Hospital for Special Surgery Knee HSS pain score improved in intervention group compared to control but not statistically significant. Sleep apnoea (1:0), manipulation (1:0), deep infection (0:1), intestinal pseudo-obstruction (0:1). |
| D'Lima et al,[73] 1996 Physical therapy USA Date not specified | Rheumatoid arthritis or osteoarthritis n=20 (10:10) 69.0 years (60%) | Exercise (strengthening, range of motion). | Hospital. 1-on-1 programme 45 min sessions, 3 times a week for 18 weeks commencing 6 weeks before surgery. | One meeting with physical therapist | 24 and 48 weeks. No losses to follow-up reported. High risk of bias: small numbers of patients; baseline demographic differences. | Hospital for Special Surgery Knee Rating pain score improved in intervention group compared to control but not statistically significant atrial fibrillation (1:0), paroxysmal tachycardia (1:0), deep infection (0:1), intestinal pseudo-obstruction (0:1). |

**Table 1** Continued

| Publication Location Date of study | Indication Number randomised (intervention: control) Mean age (%female) | Primary focus of intervention | Study setting Timing, duration and intensity | Control group care | Longest follow-up after surgery Losses to follow-up (intervention: control) risk of bias | Outcomes |
|---|---|---|---|---|---|---|
| Fernandes et al,[75] 2017 Denmark 2010–2011 | Osteoarthritis 165 (84; 81) hip or knee replacement Mean 67.9 (SD 8.6); 66.9 (8.3) (56%) | Exercise (neuromuscular focusing on lower extremity muscular control and quality of movement). | Hospital: group based, physiotherapist led Mean 13.1, 1 hour sessions twice per week during 8 weeks before surgery. | Standard care | 61 weeks. 2; 5. Low risk of bias. | Mean differences at 61 weeks favoured the exercise group for all KOOS subscales, but only significantly for quality of life subscale. Disaggregated long-term data for knee replacement acquired by personal contact 0; two deep infection. |
| Huber et al,[76] 2015 Switzerland 2009–2012 | Osteoarthritis n=45 (22:23) 70.4 years (47%) | Exercise (neuromuscular including functional and strengthening). | Hospital: group based eight or more sessions commencing 6–12 weeks before surgery. | All patients attended 'knee school' on three occasions from about 4 weeks before surgery | 12 months 9 (5:4) lost to follow-up. Low risk of bias | No difference in KOOS pain or SF-36 bodily pain between groups at 12 months one intervention patient had a joint specific adverse event. |
| Matassi et al,[77] 2014 Belgium 2005–2006 | Osteoarthritis n=122 (61:61) 66.5 years (48%) | Exercise (muscle strength and flexibility). | Home 5 days per week for 6 weeks commencing 6 weeks before surgery. | Maintained regular activities | 1 year. No losses to follow-up reported. High risk of bias due to selective outcome reporting; outcome also not patient reported. | No significant effect of exercise on the Knee Society Clinical Rating System knee score or patient function score Manipulation (5:3). In the intervention group one patient developed ipsilateral adductor tendinitis and one patient stopped exercises due to increased knee pain. |
| Rooks et al,[79] 2006 USA 2001–2003 | Osteoarthritis n=45 (22:23) 67.0 years (53%) | Exercise (strengthening, flexibility, cardiovascular, pool exercises). | Community fitness facility three times per week for 6 weeks 30–60 min each increasing intensity from 4 to 6 weeks. | Two education mailings and three telephone calls | 26 weeks. 16 (8:8) lost to follow-up. High risk of bias: large loss to follow-up. | Statistically significant greater improvement in SF-36 bodily pain at 26 weeks (p<0.05) but no difference for WOMAC pain. Perioperative complications (3:4). No serious adverse events in either group requiring hospital admission. |

Continued

**Table 1** Continued

| Publication Location Date of study | Indication Number randomised (intervention: control) Mean age (%female) | Primary focus of intervention | Study setting Timing, duration and intensity | Control group care | Longest follow-up after surgery Losses to follow-up (intervention: control) risk of bias | Outcomes |
|---|---|---|---|---|---|---|
| Tungtrongjit et al,[78] 2012 Thailand 2011–2012 | Osteoarthritis n=68 (30:30 in analyses) 64.5 years (17%) | Exercise (quadriceps strengthening). | Home three times per day for 3 weeks commencing 3 weeks before surgery. | Asked to continue normal activities | 6 months eight patients excluded from analyses due to adverse events and loss to follow-up. Possible risk of bias: unclear sequence generation; data provided only on patients followed up. | WOMAC pain improved in intervention compared with controls (p=0.029) but not VAS pain (p=0.137). Across all randomised patients there was one surgical wound infection, one postoperative knee trauma and two cases of postoperative wound dehiscence. |

EQ-5D, European Quality of Life-5 Dimensions; KOOS, Knee disability and Osteoarthritis Outcome Score; SF-36, Short Form 36; VAS, Visual Analogue Scale; WOMAC, Western Ontario and McMaster Universities Osteoarthritis Index.

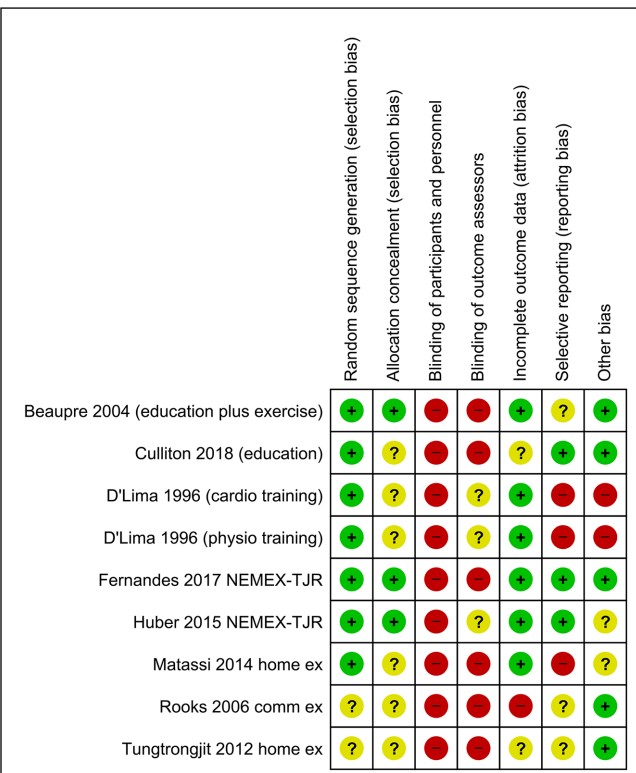

**Figure 2** Risk of bias assessments.

neuromuscular exercises combined with biomechanical training. One study included a series of pool-based exercises as well as standard indoor exercises.[79] The majority of exercise-based interventions required participants to travel to clinics or community-based exercise centres but two[77 78] involved exclusively home-based exercise. Staff delivering interventions were, in the majority of cases, physiotherapists; exceptions were one trial in which the intervention was delivered by an exercise physiologist[73] and one in which it was unclear who delivered the intervention.[77]

Participants in the study evaluating both exercise and education received a single session of education (reinforced by provision of a written materials) concerning crutch walking on level ground and stairs, bed mobility and transfers, and postoperative range of motion routines.[74] Those in the study focusing solely on education[80] received a 31-page guide as well as access to an online learning tool during their preadmission consultation, followed by subsequent reminders of their access. The customised e-learning tool included 32 brief (1–2 min) professionally filmed educational videos presented by surgeons, therapists or previous recipients of TKRs, in which expectations related to pain, functional outcomes, limitations and restrictions were addressed, as well as animations allowing them to visualise the surgery itself.

### Intensity, duration, measures of adherence/compliance

Exercise interventions were delivered for between 3 and 12 weeks, 2–7 times a week. The shortest duration of intervention was a 3-week home-based intervention[78] which featured the most intense dose (three sessions per

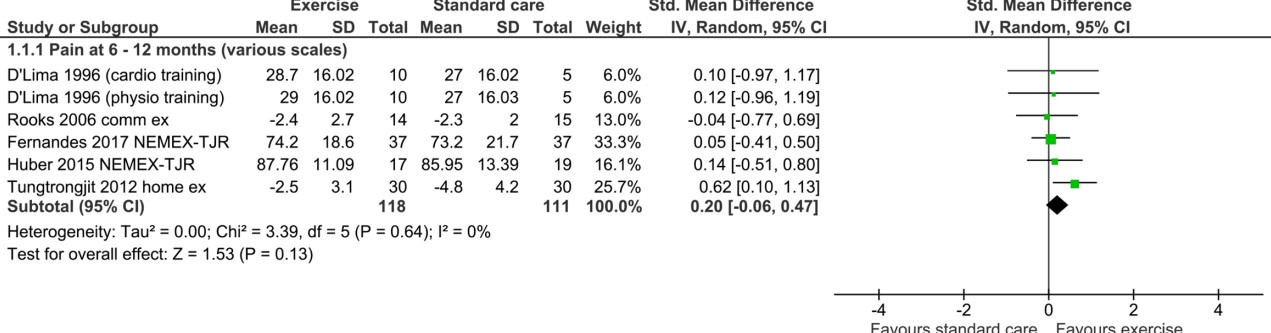

**Figure 3** Forest plot: exercise.

day, 7 days a week). The longest period of intervention (8–12 weeks) took place in the two trials[75 76] delivering the clinic-based NEMEX-TJR intervention.[81] Here, variability in duration could be determined by how close to the surgery a patient was at time of recruitment.

With regard to exercise interventions conducted within clinics or at home, investigators typically assessed adherence via logbooks[74–77] recording attendance and sometimes details of achievement and effort. One study of a home-based intervention without formal reporting of adherence featured daily telephone calls to participants[78]; two clinic-based interventions[73] reported that all participants completed all sessions. The study focussing solely on education[80] assessed the number of overall 'hits' to different educational videos concerning different aspects of the TKR pathway.

### Comparator groups

Among the trials assessing the effects of exercise or exercise combined with education, detail and nature of comparator groups varied. Five trials described comparator groups as 'usual care'. For three trials the descriptions of usual care were brief but indicated that participants continued with regular activities until surgery.[74 77 78] In one trial of cardiovascular exercise, usual care was described as participants having a single meeting with a physiotherapist before their operation for 45 minutes and being provided with printed material about postoperative exercise regimens.[73] In another trial, usual care involved all participants being offered a 'standard preoperative educational package' comprising written information on the operation, 'expected postoperative progression' and recommended exercises, and a

3-hour in-clinic information session led by health professionals 1 week prior to surgery.[75]

In the two other studies assessing exercise, the comparator groups were designed as attentional controls. In one, participants were sent information on joint replacement, recommended home modifications and on preparing for surgery, reinforced by three telephone calls.[79] The most intensive control condition was 'knee school' offered alongside an exercise intervention but also offered to the control group participants.[76] The knee school included formal teaching and peer discussions, was led by an experienced and specially trained physiotherapist over 3 weekly individual or group sessions. Participants were provided with information on anatomy (including photos, diagrams and models), recommended activities after surgery, postoperative pain management and rehabilitation.

In the trial evaluating online education, the control group were provided with a 31-page copy of 'My Guide to Total Knee Joint Replacement'.[80]

### Outcomes and outcome data

Six studies reported a joint-specific or condition-specific patient reported pain outcome measure.[74–76 78–80] In two publications (one reporting data for two separate comparisons), the outcomes reported were surgeon-assessed.[73 77]

The latest pain assessment was reported at about 6 months[73 74 76 77] or 12 months after surgery.[78–80] Measures used were WOMAC pain,[74 78] KOOS pain,[76 80] SF-36 bodily pain,[74 76 80] VAS pain,[78] HSS pain score[73] and KSS.[77]

We made robust attempts to obtain data through contact with investigators from studies in which data were provided within graphs without SD,[73 76] in tables

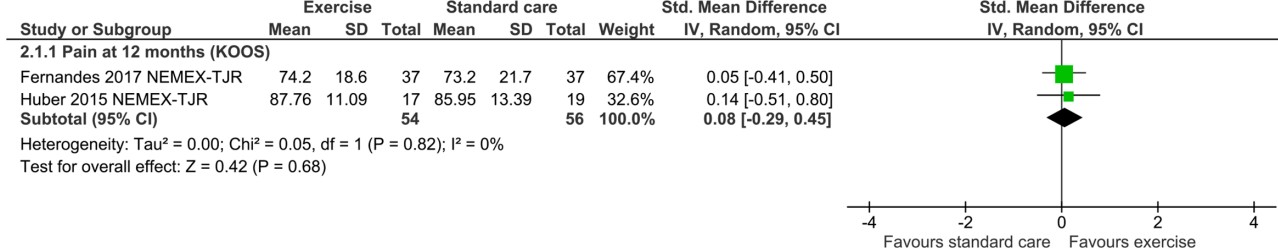

**Figure 4** Forest plot: exercise sensitivity analysis.

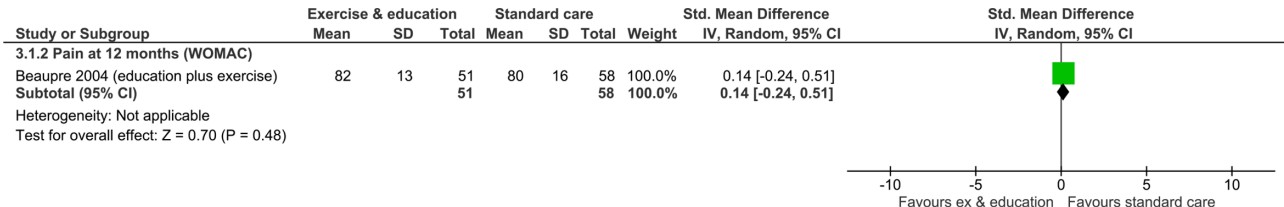

**Figure 5** Forest plot: exercise and education.

with means but without SD,[80] as 'non significant' only[77] or as part of aggregated hip and knee data.[75] In three studies, investigators generously provided relevant data in the form of group means and SDs and these data were used.[75 76 80] Following information that data for one study had been permanently lost,[73] we measured WOMAC pain data from a graph and, in the absence of SD, used values from a cohort of patients with the same outcome measure reported after TKR but converted from a 50-point to a 30-point scale.[82] The first author of an exercise-based study which had reported pain findings as simply 'n.s.' (non-significant), responded to our enquiry to indicate that they could no longer access study data.

### Risk of bias
Four studies were assessed as being at an overall low risk of bias.[74 76 77 80] In other studies, we had concerns about potential biases due to differences in baseline patient characteristics in two comparisons[73] and in one study each, large losses to follow up[79] and issues regarding sequence generation.[78]

### Effects of intervention: exercise alone
#### Pain at 6 months or longer
Combined results of six eligible interventions reported in five studies[73 75 76 78 80] indicate no clear effect of intervention on long-term pain (SMD 0.20, 95% CI −0.06 to 0.47; participants=229). There was no heterogeneity across studies, $\chi^2$=3.39, $I^2$=0% (figure 3).

In a sensitivity analysis restricted to two studies with low risk of bias,[74 76] group results provided even less suggestion of difference between groups (SMD 0.08, 95% CI −0.29 to 0.45; participants=110; $I^2$=0%) (see figure 4).

Results for one study (n=122) within this category for which no data were suitable for meta-analysis are in line with the results above, with 'non significant'[77] findings.

### Effects of intervention: exercise combined with education
#### Pain at 6 months or longer
Results of a single study of exercise with the additional component of one session of education (n=109 followed up) suggest no clear difference between intervention and standard care on long-term pain as assessed by the WOMAC pain scale (MD 2.00, 95% CI −3.45 to 7.45)[74] (figure 5).

### Effects of intervention: education alone
#### Pain at 6 months or longer
Results of a single study of education provided by an e-learning tool (n=319) suggest no clear difference between intervention and standard care in terms of long-term pain as assessed on the KOOS (MD −2.55, 95% CI −6.35 to 1.24)[80] (figure 6).

### Adverse events
Perisurgical and postsurgical complications including infections were assessed in most studies; see table 1. Where reported, these do not appear to have differed between groups. Two studies reported presurgical events we judged likely to be intervention-related[76 77]; in each case, participants in an exercise intervention group reported increased pain. No data for this outcome were reported in the study focussing on education alone.[80]

### DISCUSSION
#### Summary of main findings
In people with osteoarthritis receiving TKR, the potential value of interventions to manage preoperative risk factors or to target care has frequently been assessed, most often focussing on interventions this review has identified (exercise or 'prehabilitation' and/or education with the aim of managing expectations). We found low-quality to moderate-quality evidence for no effect on our primary

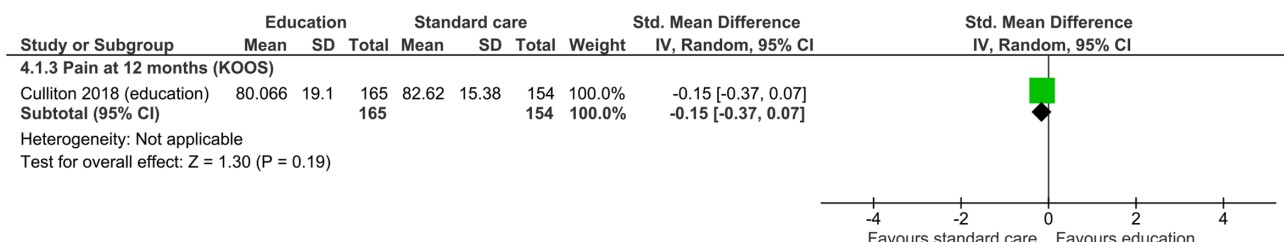

**Figure 6** Forest plot: education alone.

**Table 2** Summary of findings tables

Summary of findings: exercise alone compared with standard care for knee replacement (long-term outcome only)

Patient or population: adult patients scheduled for total knee replacement
Setting: clinic-supervised exercise (five studies); home-based exercise (two studies)
Intervention: 3–12 weeks' exercise conducted in the presurgical period
Comparison: standard care

| Outcome<br>No of participants<br>(studies) | Relative effect (95% CI) | Illustrative comparative risks (95% CI) | | Certainty | Comments |
|---|---|---|---|---|---|
| **Long-term pain**<br>Assessed with: various scales (WOMAC*, KOOS†, HSSK‡)<br>Follow-up: range 6–12 months<br>No of participants in meta-analysis: 229 (6 RCTs)<br>No of participants not in meta-analysis: 122 (1 RCT) | **Meta-analysis:**<br>SMD 0.2 higher (−0.06 to 0.47)<br>**Narrative**<br>'Non-significant difference' | Mean KOOS scores in the untreated group ranged from 73.2 to 85.95 in the control group; mean HSSK scores in the control group were 27; mean WOMAC scores in control group were −2.3 to −4.8 | The mean level of pain after exercise was 0.20 SD lower (−0.06 to 0.47 lower). | ⊕⊕⊕◯<br>MODERATE§¶ | There was no clear evidence of a difference between exercise and standard care for long-term pain either from data from the meta-analysis or from a comparatively large study not included within the meta-analysis.<br>We rescaled the SMD finding to a commonly used instrument in the field – the KOOS, using estimations based on the mean of a representative study within the analysis (Huber 2015) and our pooled SMD of 0.02 (−0.06 to 0.47 higher). This indicates that exercise led to a mean increase of 2.7 units/points on the KOOS (95% CI −0.8 to 6.29) in the treated group.<br>Sensitivity analysis of two studies at low risk of bias suggested a smaller effect (mean increase of 1.46 points on the KOOS (95% CI-4.58 to 7.50) |
| **Adverse events** | See comment | See comment | See comment | ⊕⊕◯◯<br>LOW§** | Two studies reported treatment-related adverse events (increase in pain) separately from the main pain outcomes, which occurred during the period of intervention. Information on perioperative complications, for example, superficial and deep infections, were reported in all studies. Events were few and attribution to treatment status difficult |

**GRADE: Working group grades of evidence**
**High certainty:** We are very confident that the true effect lies close to that of the estimate of the effect.
**Moderate certainty:** We are moderately confident in the effect estimate: The true effect is likely to be close to the estimate of the effect, but there is a possibility that it is substantially different.
**Low certainty:** Our confidence in the effect estimate is limited: The true effect may be substantially different from the estimate of the effect.
**Very low certainty:** We have very little confidence in the effect estimate: The true effect is likely to be substantially different from the estimate of effect.

Continued

**Table 2** Continued

**Summary of findings: exercise alone compared with standard care for knee replacement (long-term outcome only)**

**Patient or population:** adult patients scheduled for total knee replacement
**Setting:** clinic-supervised exercise (five studies); home-based exercise (two studies)
**Intervention:** 3–12 weeks' exercise conducted in the presurgical period
**Comparison:** standard care

| Outcome No of participants (studies) | Relative effect (95% CI) | Illustrative comparative risks (95% CI) | Certainty | Comments |
|---|---|---|---|---|
| **Long-term pain** Assessed with: WOMAC* scale Follow-up: 12 months No of participants: 109 (1 RCT) | **Single study results:** MD **2 units higher** (3.45 lower to 7.45 higher) | | | |

*WOMAC (Bellamy et al 1988).[55] The original pain scale runs from 0 to 20 and a higher number indicates greater pain. There are several modifications. Some trialists also reverse the polarity and/or transform responses into a 0–100 scale where a high number indicates less pain. We standardised to this latter method as it was most common and was similar to the KOOS and the HSSK.

†KOOS (Roos et al 1998).[56] The scale ranges from 0 to 100 and a higher number indicates less pain.

‡HSSK (Insall et al 1976).[58] Pain is measured on a scale of 0–30, with a high number indicating less pain.

§Downgraded due to risk of bias associated with lack of blinding of both personnel and participants in all studies also concerns about sequence generation/allocation concealment in three studies.

¶We did not downgrade for imprecision as findings of the one large study not included in metaanalysis were very similar to the six studies included in meta analysis.

**We downgraded the quality of evidence for study limitations, in particular imprecision: estimate based on few events.

GRADE, Grading of Recommendations Assessment, Development and Evaluation; HSSK, Hospital Society Score; KOOS, Knee Injury and Osteoarthritis Outcome Score; MD, mean difference; SMD, standardised mean difference; WOMAC, Western Ontario and McMaster Universities Osteoarthritis Index.

**Summary of findings: exercise and education compared with standard care for knee replacement (long-term outcome only)**

**Patient or population:** Adult patients scheduled for total knee replacement
**Setting:** Clinic-supervised exercise Clinics (one study)
**Intervention:** 1 session of education combined with 4 weeks' exercise in the preoperative period
**Comparison:** TAU

| Outcome No of participants (studies) | Relative effect (95% CI) | Illustrative comparative risks (95% CI) | Certainty | Comments |
|---|---|---|---|---|
| | | Mean score of untreated control group was 80 on the WOMAC (SD=16) | Exercise and education resulted in a mean score of 82 on the WOMAC (SD=13) | ⊕⊕○○ LOW†‡§ |
| **Adverse events** | See comment | See comment | ⊕⊕⊕○ VERY LOW†‡§¶ | General complications, 'similarly dispersed across both groups': Pulmonary embolism (2:2), DVT (3:6), superficial infection (2:3), deep infection (1:0), manipulation (1:2). |

**GRADE Working Group grades of evidence**
**High certainty:** We are very confident that the true effect lies close to that of the estimate of the effect.
**Moderate certainty:** We are moderately confident in the effect estimate: The true effect is likely to be close to the estimate of the effect, but there is a possibility that it is substantially different.
**Low certainty:** Our confidence in the effect estimate is limited: The true effect may be substantially different from the estimate of the effect.
**Very low certainty:** We have very little confidence in the effect estimate: The true effect is likely to be substantially different from the estimate of effect.

Continued

**Table 2** Continued

## Summary of findings: exercise and education compared with standard care for knee replacement (long-term outcome only)

**Patient or population: Adult patients scheduled for total knee replacement**
Setting: Clinic-supervised exercise Clinics (one study)
Intervention: 1 session of education combined with 4 weeks' exercise in the preoperative period
Comparison: TAU

| Outcome No of participants (studies) | Relative effect (95% CI) | Illustrative comparative risks (95% CI) | Certainty | Comments |
|---|---|---|---|---|

*WOMAC (Bellamy et al, 1988).[55] The original pain scale runs from 0 to 20 and a higher number indicates a greater pain. There are several modifications. Some trialists also reverse the polarity and/or transform responses into a 0–100 scale where a high number indicates less pain. We standardised to this latter method as it was most common and was similar to the KOOS and the HSSK.
†Downgraded due to risk of bias due to lack of blinding of participants or personnel.
‡Downgraded for imprecision due to being one study with less than an optimal information sample size.
§There were too few studies to reliably assess risk of publication bias or other sources of small study bias.
¶We downgraded the quality of evidence for study limitations, in particular imprecision: estimate based on few events.
GRADE, Grading of Recommendations Assessment, Development and Evaluation; HSSK, Hospital Society Score; KOOS, Knee Injury and Osteoarthritis Outcome Score; MD, mean difference; TAU, treatment as usual; WOMAC, Western Ontario and McMaster Universities Osteoarthritis Index.

## Summary of findings: Education compared with standard care for knee replacement (long-term outcome only)

**Patient or population: Adult patients scheduled for total knee replacement**
Setting: Clinic followed by online access (one study)
Intervention: Access to online elearning tool with active demonstration at presurgical visit followed by email prompts
Comparison: TAU

| Outcome No of participants (studies) | Relative effect (95% CI) | Illustrative comparative risks (95% CI) | Certainty | Comments |
|---|---|---|---|---|
| **Long-term pain** Assessed with: KOOS scale* Follow-up: 12 months No of participants: 319 (1 RCT) | **Single study results:** MD **2.55 lower** (6.35 lower to 1.24 higher) | Mean score of untreated control group was 82.62 on the KOOS (SD=15.4) Exercise and education resulted in a decrease on the KOOS, leading to a mean score in the treated group of 80.01(SD=19) | ⊕⊕⊕◯ MODERATE†‡ | |
| Adverse events | See comment | See comment | | See comment No data on adverse events appear to have been sought within the one study within this comparison. |

**GRADE Working Group grades of evidence**
**High certainty:** We are very confident that the true effect lies close to that of the estimate of the effect.
**Moderate certainty:** We are moderately confident in the effect estimate: The true effect is likely to be close to the estimate of the effect, but there is a possibility that it is substantially different.
**Low certainty:** Our confidence in the effect estimate is limited: The true effect may be substantially different from the estimate of the effect.
**Very low certainty:** We have very little confidence in the effect estimate: The true effect is likely to be substantially different from the estimate of effect.

*KOOS (Roos et al, 1998).[56] The scale ranges from 0 to 100 and a higher number indicates less pain.
†Downgraded due to risk of bias due to lack of blinding of participants or personnel and high loss to followup (24%).
‡There were too few studies to reliably assess risk of publication bias or other sources of small study bias.
GRADE, Grading of Recommendations Assessment, Development and Evaluation; KOOS, Knee Injury and Osteoarthritis Outcome Score; MD, mean difference.

outcome (pain assessed at 6 months or longer) of either exercise, education or education combined with exercise, and we identified no eligible studies assessing the effects of any other eligible preoperative interventions (eg, weight loss programmes or smoking cessation).

## Strengths and limitations of this study

With broad inclusion criteria, we aimed to identify diverse interventions conducted in the preoperative period. Important issues that our review benefits from are preregistration of the study protocol to limit reporting bias, conduct of study according to PRISMA guidelines and comprehensive up to date searches of a range of appropriate literature databases. Experienced systematic reviewers undertook screening of articles, data extraction and risk of bias assessment in duplicate. Author contact was extensive and with the help of study investigators, we included data from studies which had been excluded from previous meta-analyses or had been included with estimates for variances. Furthermore, use of GRADE methodology allowed us to assess the overall quality of the evidence.

A limitation of our review concerns our focus on prevention of long-term pain only. This decision was based on guidelines which emphasise that osteoarthritic pain which is not controlled by conservative treatments is a primary reason that people undergo TKR.[83] Prevention of long-term pain is a key determinant of patient satisfaction with TKR.[84] Although the primary outcome of many studies we identified was function, pain severity was an important secondary outcome. We cannot draw conclusions on the value of the interventions we have examined from the point of view of (for example) the benefit to individuals or society of quicker short-term recovery, as proposed by some trialists.[75]

The secondary outcome of this review was adverse events, but reporting was lacking. The poor quality of adverse event reporting in surgical trials is recognised,[85] and standardisation is required to improve quality and reduce heterogeneity, particularly in orthopaedics. Although patient reported outcomes are frequently considered to be at high risk of bias, we considered them to be appropriate in this review where questionnaires were completed over 6 months after a presurgical intervention.

We also excluded studies in which the intervention continued beyond the period of the TKR procedure; a separate review considering studies including interventions which combined prehabilitation and post-TKR rehabilitation may well be justified.

## This review in context

This is the first review in nearly a decade[35] which takes as its focus the preoperative period itself, rather than aiming at a class of interventions; the evidence base, however, remains limited.

Previous reviews have focused on outcomes relating to preoperative function and pain, the in-hospital experience and long-term recovery.[10 12–46] Review conclusions vary based on the outcomes and follow-up times. In general, those concentrating on short-term function and outcomes related to discharge are more likely to report early patient benefit and shorter hospital stay for patients receiving presurgical exercise and education compared with controls. Those considering longer-term postoperative outcomes show little difference in outcomes between randomised groups. One review which evaluated the effects of 'prehabilitation' across *all* surgical patients concluded that '*prehabilitation studies are not recommended in patients with osteoarthritis for whom arthroplasty is planned*'.[14] Subsequently, among the most comprehensive of systematic reviews conducted in arthroplasty alone is that by Wang and colleagues.[46] They concluded that the effects of prehabilitation on long-term and indeed many short-term outcomes were too small to be considered clinically important.

Our review adds to the existing evidence base as it includes two trials not included in other reviews[76 80] and previously unreported data from an older trial[75] which was acquired from study authors. Inclusion of data from these three trials in our meta-analysis did not support use of preoperative exercise or education interventions for the prevention of chronic pain but evidence supporting this was of low to moderate quality. This is in line with previous review conclusions.[45 46] Furthermore, a range of preoperative interventions have not been evaluated in randomised trials with long-term follow-up for pain outcome.

## Implications for practice and research

There are several possible explanations for the observed lack of a convincing effect of preoperative interventions on long-term pain after TKR reported within this review. All preoperative interventions conducted within an undifferentiated general population may appear ineffective in the long-term, as for many patients, TKR surgery is highly effective in reducing long-term pain. To detect an effect of intervention for the small but important minority of patients for whom TKR is not highly effective in reducing long-term pain, studies may need either to be much larger, or more focused on particular patient subgroups. Despite the fact that patients report that appropriate and realistic support before surgery can help them achieve a long-term positive experience of surgery,[86] it is acknowledged that it is difficult to recruit patients to randomised controlled trials before surgery and participants in existing trials may have been highly selected. Patients may have exhausted conservative strategies before surgery and, in evaluating new approaches, there may be an unwillingness among patients to participate in randomised trials.[10] Intervention content may have focused on improving preoperative physical function and preparing patients for their hospital admission and recovery without specific reference or 'flags' to help identify participants whose recovery did not follow the norm.

An aspect of presurgical health with a possible link with chronic pain is psychological distress.[87] As an example, pain catastrophising has been implicated as a risk factor for chronic pain.[88] However, a recent trial where intervention began in the presurgical period and extended well beyond it found that pain coping skills training for

people with high pain catastrophising did not improve WOMAC pain scores at 12 months after TKR.[89] Other aspects of preoperative psychological distress are associated with long-term pain after TKR and evaluation of treatment strategies merits investigation.[87] For other preoperative strategies, such as smoking cessation, weight loss and comorbidity management, there may be tenuous, if any, mechanisms to link with long-term pain outcomes. However, they share a common aim in preparing patients for surgery and preventing adverse events that may limit the potential benefit of knee replacement, so we believe it right to advocate that studies investigating their effects incorporate long-term follow-up.

## CONCLUSION

Presurgical interventions have been evaluated in patients receiving TKR. We found low to moderate-quality evidence to suggest that exercise is not effective in preventing long-term pain in adults scheduled to receive TKR, and it is difficult to recommend further research except of novel approaches and/or in specific populations. The evidence base for education and exercise combined with education was highly limited. However, studies of preoperative interventions with long-term follow-up after surgery are feasible and adequately powered, randomised controlled trials, planned and reported according to Consolidated Standards of Reporting Trials standards, should be conducted to identify clinically effective preoperative treatments to help patients achieve a long-term pain-free outcome after knee replacement.

**Acknowledgements** We gratefully acknowledge unpublished outcome data from the following investigators: Dr Sharon Culliton, Dr Erika Huber and Dr Allan Villadsen. We also are thankful for confirmation concerning other study data from Professor Darryl D'Lima and Dr Fabrizio Matassi. We are also thankful for translation of Yunfei Li, University of Bristol, for his help in translating a paper from Chinese.

**Contributors** All authors contributed to the concept and design of the study. JD, ADB and VW contributed to the acquisition and analysis of data. JD and ADB drafted the article and VW, RG-H and AWB revised it critically for important intellectual content. JD and ADB take responsibility for the integrity of the work as a whole, from inception to finished article.

**Funding** This study is funded by the National Institute for Health Research (NIHR) [NIHR Programme Grant for Applied Research (Grant Reference Number RP-PG-0613–20001)]. This study was also supported by the NIHR Biomedical Research Centre at the University Hospitals Bristol NHS Foundation Trust and the University of Bristol. The views expressed in this publication are those of the authors and not necessarily those of the NHS, the National Institute for Health Research or the Department of Health and Social Care.

**Disclaimer** The views expressed are those of the authors and not necessarily those of the NHS, the NIHR or the Department of Health. The funder had no involvement in the study design, data collection, data analysis, interpretation of data or writing of the manuscript.

**Competing interests** Outside of this work, VW and AWB are co-applicants on an institutional grant from Stryker for a study evaluating the outcomes of the Triathlon knee replacement.

**Patient consent for publication** Not required.

**Provenance and peer review** Not commissioned; externally peer reviewed.

**Data availability statement** No data are available. All data relevant to the study are included in the article or uploaded as supplementary information.

**ORCID iDs**
Jane Dennis http://orcid.org/0000-0001-9718-2653
Andrew David Beswick http://orcid.org/0000-0002-7032-7514

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
