## [Reviewer comments · BMJ Open]

ARTICLE DETAILS

TITLE (PROVISIONAL)	The effects of pre-surgical interventions on chronic pain after total knee replacement: A systematic review and meta-analysis of randomised controlled trials
AUTHORS	Dennis, Jane; Wylde, Vikki; Goberman-Hill, Rachael; Blom, AW; Beswick, Andrew

VERSION 1 – REVIEW

REVIEWER	Gwyn Lewis Auckland University of Technology
REVIEW RETURNED	15-Aug-2019

GENERAL COMMENTS	I think the authors need to better acknowledge previous research in the area. It is stated in the Discussion that “This is the first review in nearly a decade which takes as its focus the preoperative period itself”, when the authors go on to cite two recent reviews that examine preoperative interventions for TKR (Wang et al., 2016; Cabilan et al., 2015). There is at least one other similar review (Chemham & Shanmugam, 2016). Given this, the authors should be clearer on what is novel in this review and what it adds to the current literature. There should also be some mention of these reviews and their findings in the Introduction. Currently, there is a vague statement that the impact of presurgical interventions on long-term pain remains to be established, yet there are at least three previous systematic reviews that have looked at this. I found the formatting of the Results section difficult to follow. There seemed to be an overview first, then separate sections based on (presumably?) intervention types, although this is not clear. Given the lack of range of interventions, the overview information seemed to be repeated in the following section on exercise and education. I also wasn't sure why exercise and education were grouped together as they are quite separate entities to me. The grouping together of study setting, sample sizes and foci (?) also was difficult to follow. I think the heading in this section could be better thought out. The description of control/comparison interventions is missing and it is difficult to make conclusions about the interventions with this. I am not sure why the sections on drugs/supplements, smoking cessation, or bacterial screening sections are present. If there were no studies using these interventions that met the inclusion criteria, they do not need to be reported upon. There are quite a few grammatical errors throughout the manuscript and often the referencing format is incorrect.
--

	Abstract, line 42. There were 5 studies evaluating exercise? Strengths and limitations, line 9. Both “long-term” and “chronic” are not required. The meaning of the 3rd point is not clear. Page 4, lines 16-17. I don’t see the relevance of this sentence in relation to the rest of the paragraph. Page 4, lines 33-37. I am not clear how these potential mechanisms relate to the risk factors. Come up again in Discussion. Page 5, lines 37-39. This is not a proper sentence. Page 6, line 47. There is a word missing. Page 8, line 12. What is meant by “At protocol stage”? Page 8, final sentence. Grammar in this sentence needs improving. Page 9, line 7. It is not clear what “foci” relates to. Page 9, lines 16-19. Grammar in this sentence needs improving. Page 9, 55-58. I suggest further grouping of interventions – range of movement, stretching, flexibility appear to be equivalent. Were pool-based activities aerobic/strengthening? What is neuromuscular exercise? Page 10, lines 5. Be specific about the number of studies that had physiotherapist supervision. Page 10, line 30. What is NEMEX? Page 12, final sentence. There were no studies found related to other interventions that met the inclusion criteria. This should be clarified in this sentence. Page 13, line 22. There is a word missing. Page 13, lines 32-35. Studies were excluded where the INTERVENTION continued beyond the TKR procedure.
--	--

REVIEWER	Amin Mohamadi Harvard Medical School, USA
REVIEW RETURNED	13-Sep-2019

GENERAL COMMENTS	Thank you for invitation on statistical review on this systematic review and meta-analysis of randomized clinical trials on pre-surgical interventions on chronic pain after total knee replacement. The authors electronically searched Medline, Embase, Cinahl, Cochrane library and PsycInfo and citation of included studies were checked for potential missing studies. The risk of bias was assessed using Cochrane tool and GRADE approach was used to evaluate strength of evidence. Outcomes included patient-reported pain intensity but this could be evaluated by number of different tools. However, the appropriate section of each tool that assesses subjective pain was employed for statistical analysis. The statistical analysis seems appropriate, the results are well organized and overall the study is well conducted and written. I have one comment: Authors mentioned they have used random effect meta-analysis however the I2 statistic is 0% for exercise intervention. I2 is not presented for exercise combined with education or education alone as interventions. Please present I2 statistic for all of quantitative analyses and if in fact fixed effect analysis has been used, revise that.
---

VERSION 1 – AUTHOR RESPONSE

Reviewer: 1

I think the authors need to better acknowledge previous research in the area. It is stated in the Discussion that “This is the first review in nearly a decade which takes as its focus the preoperative period itself”, when the authors go on to cite two recent reviews that examine preoperative interventions for TKR (Wang et al., 2016; Cabilan et al., 2015). There is at least one other similar review (Chemham & Shanmugam, 2016). Given this, the authors should be clearer on what is novel in this review and what it adds to the current literature. There should also be some mention of these reviews and their findings in the Introduction. Currently, there is a vague statement that the impact of presurgical interventions on long-term pain remains to be established, yet there are at least three previous systematic reviews that have looked at this.

Thank you.

There have been numerous reviews on the topic of pre-surgical interventions and in the Discussion, we cited 19 reviews including the reviews highlighted in your comment.

We have now described the reviews in the Introduction section more carefully with reference to search dates and other features that help contextualise our work better.

We note in the Discussion that our review includes data from trials that were not included in previous reviews, thus highlighting what our review adds to the existing literature.

I found the formatting of the Results section difficult to follow. There seemed to be an overview first, then separate sections based on (presumably?) intervention types, although this is not clear. Given the lack of range of interventions, the overview information seemed to be repeated in the following section on exercise and education. I also wasn't sure why exercise and education were grouped together as they are quite separate entities to me. The grouping together of study setting, sample sizes and foci (?) also was difficult to follow. I think the heading in this section could be better thought out. The description of control/comparison interventions is missing and it is difficult to make conclusions about the interventions with this.	We would like to thank you for these helpful suggestions for improving the formatting of the results section. To address your comments, we have made the following amendments:  (1) We have reorganised based on the simpler lines of the PRISMA flowchart (now submitted) and (2) We have omitted the sections on drugs/supplements, smoking cessation, and bacterial screening sections which featured in full paragraphs previously, and mention them now only as studies which met criteria for long-term follow-up but lacked an assessment of pain (page 9). (3) We have included a section describing the care provided to the control group (pp 11-12).
I am not sure why the sections on drugs/supplements, smoking cessation, or bacterial screening sections are present. If there were no studies using these interventions that met the inclusion criteria, they do not need to be reported upon.	
There are quite a few grammatical errors throughout the manuscript and often the referencing format is incorrect.	We apologise for this and we have now addressed grammatical errors and referencing format throughout the manuscript.
Abstract, line 42. There were 5 studies evaluating exercise?	There were indeed 5 studies but one included two eligible comparisons. Therefore, in meta-analysis it is treated as two. We have clarified this in the Abstract.
Strengths and limitations, line 9. Both "long-term" and "chronic" are not required. The meaning of the 3rd point is not clear.	Thank you. We have removed the word 'long-term'. We have also sought to clarify the third point as follows: "We only included studies that completed intervention delivery during the pre-operative period and did not include studies that evaluated interventions that began in the pre-operative period but extended into the peri-operative and post-operative period."
Page 4, lines 16-17. I don't see the relevance of this sentence in relation to the rest of the paragraph.	Thank you, we agree that this sentence is superfluous within this paragraph, and we have therefore deleted: "Knee pain at rest and during walking are important indications which guide decisions for joint replacement in people with osteoarthritis".
Page 4, lines 33-37. I am not clear how these potential mechanisms relate to the	We removed the line mentioned by the reviewer (beginning "Associations between risk factors...") and

risk factors. Comes up again in Discussion.	also removed related text in the Discussion (see page 16).
Page 5, lines 37-39. This is not a proper sentence.	We have corrected this error. This sentence on p 5 now reads “Eligible participants were adults on the waiting lists for knee replacement, up until the point of admission for surgery
Page 6, line 47. There is a word missing.	We have inserted the word “trials” (this section now appears on p 7).
Page 8, line 12. What is meant by “At protocol stage”?	We have inserted a reference to our PROSPERO registration where we use the phrasing “at protocol stage”. This provides a clearer link to the protocol which is moreover (as above) now updated.
Page 8, final sentence. Grammar in this sentence needs improving.	We have reviewed and amended this sentence to improve the grammar (top of page 9).
Page 9, line 7. It is not clear what “foci” relates to.	We have changed this to “focus of intervention” and also changed “sample sizes” to “sample size” for consistency (page 9).
Page 9, lines 16-19. Grammar in this sentence needs improving.	We have amended this sentence to improve the grammar (p 9).
Page 9, 55-58. I suggest further grouping of interventions – range of movement, stretching, flexibility appear to be equivalent. Were pool-based activities aerobic/strengthening? What is neuromuscular exercise?	Thank you, this is a particularly helpful comment. We have clarified the writing here and have removed reference to ‘pool based’ as we agree that the term is vague and it is preferable to rely on the targets of each intervention (strength, flexibility, etc). See page 10.
Page 10, lines 5. Be specific about the number of studies that had physiotherapist supervision.	We have now clarified the number of studies that had physiotherapist intervention on page 10. All trials reported physiotherapy supervision with the exception of two trials. There was one trial in which the intervention was delivered by an exercise physiologist (D’Lima) and another trial in which the background of the person delivering the intervention was unclear (Matassi).
Page 10, line 30. What is NEMEX?	We now explain NEMEX in more detail. NEMEX-TJR (NEuroMuscular Exercise training program for patients with knee or hip osteoarthritis (OA) assigned for total joint replacement (TJR)), is the manualised programme used in two trials (Huber and Villadsen/Fernandes), and we have clarified this in the manuscript (page 10). We have also added the more of the relevant acronym (NEMEX-TJR not just NEMEX) and a reference (Ageberg E, Link A, Roos EM. Feasibility of neuromuscular training in patients with severe hip or knee OA: the individualized goal-based NEMEX-TJR training program. BMC Musculoskelet. Disord. 2010)

Page 12, final sentence. There were no studies found related to other interventions that met the inclusion criteria. This should be clarified in this sentence.	To clarify this we have amended this sentence by adding “and we identified no eligible studies assessing the effects of any other pre-operative interventions (e.g. weight loss programmes or smoking cessation).”
Page 13, line 22. There is a word missing.	We have corrected this sentence to read “The secondary outcome of this review was adverse events, but reporting was lacking”
Page 13, lines 32-35. Studies were excluded where the INTERVENTION continued beyond the TKR procedure.	Thank you, we have now corrected this sentence so it is unambiguous that it is the delivery of the intervention that was key (not the study itself).

Reviewer: 2

Thank you for invitation on statistical review on this systematic review and meta-analysis of randomized clinical trials on pre-surgical interventions on chronic pain after total knee replacement. The authors electronically searched Medline, Embase, Cinahl, Cochrane library and PsycInfo and citation of included studies were checked for potential missing studies. The risk of bias was assessed using Cochrane tool and GRADE approach was used to evaluate strength of evidence. Outcomes included patient-reported pain intensity but this could be evaluated by number of different tools. However, the appropriate section of each tool that assesses subjective pain was employed for statistical analysis. The statistical analysis seems appropriate, the results are well organized and overall the study is well conducted and written	Thank you for your kind comments. We are pleased that you found our analysis appropriate, the results well organised and the study well conducted and written.
--	--

I have one comment: Authors mentioned they have used random effect meta-analysis however the I ² statistic is 0% for exercise intervention. I ² is not presented for exercise combined with education or education alone as interventions. Please present I ² statistic for all of quantitative analyses and if in fact fixed effect analysis has been used, revise that.	Thank you. As described in our methods, we anticipated heterogeneity due to variation in protocols for the included interventions and the treated populations, and therefore planned a random effects model. We agree that I ² values were 0% and we have revised the abstract to add the I ² value. The Cochrane Handbook advises not to choose between effect models based on statistical tests so we have retained the existing graphs. We hope this is acceptable given that as there is no heterogeneity, these values would remain identical with a fixed effect model. This means that we cannot add I ² for education combined with exercise or for exercise alone, because no heterogeneity is possible where there is only one study.
---	--

VERSION 2 – REVIEW

REVIEWER	Gwyn Lewis Auckland Univ Technol
REVIEW RETURNED	17-Nov-2019

GENERAL COMMENTS	The authors have addressed most of my comments well. There are a few remaining queries. There needs to be some clarification of the number of articles, RCTs, and comparisons in the review. In the Abstract it is stated there were 8 RCTs with 9 comparisons. In the Results section of the main text, it states initially that there were 10 articles and 8 RCTs, although only 8 articles are cited. The following sentence then states there were 10 RCTs and 9 comparisons. However, the following sentences describe interventions for 8 studies, although there are 9 publications in Table 1. Abstract, line 41. The number of studies reported to evaluate exercise is incorrect. Page 4, line 43-46. Stating that twelve of the reviews were older than 5 years and 5 only looked at short-term outcomes doesn't provide much useful information when we don't know what "numerous" means. If 30 reviews had been conducted, this leaves a large number unaccounted for. Were these reviews searched for systematically, i.e. do the authors know they have an exhaustive list? This needs to be clarified to put these descriptions into context. Page 4, lines 54-56. This sentence doesn't provide much useful information. What were the interventions? Does "small difference" mean significant but not clinically meaningful? Which groups had less pain? What does "not durable" mean? Page 10, lines 11-13. There still appears to be some overlap in the intervention descriptions. What is the difference between range of movement, flexibility, and stretching exercises? page 13, line 40. "either" implies two options. Page 14, lines 31-34. I am not sure what this sentence means. Page 14, lines 47-49. Given that these two articles are the only differences from previous reviews, do they make any difference to the conclusions? It seems they continue to support previous research. The lower part of this paragraph discusses information that is more relevant to the strengths and weaknesses section. There is a lack of discussion on the findings of preoperative education - most of this section discusses previous reviews/studies on exercise. Page 15, lines 34-39. I don't understand what the main point of this sentence is. Exercise and education were the only interventions included in the review. Page 15, lines 41-48. I am not quite sure what the first sentence means. What is "lack of psychological support"? It should be noted
---

	that the study referred to here didn't actually manipulate pain catastrophizing, as there were no differences in this among the groups. Page 16, lines 6-10. Given that there was only 1 study that looked at education alone, would the authors not recommend further studies using education as an intervention? I am not sure what "in other areas" refers to in the following sentence.
--	---

VERSION 2 – AUTHOR RESPONSE

We thank Reviewer 1 for his thorough consideration of our revised article. We are pleased for the opportunity to address the issues raised.

R1: There needs to be some clarification of the number of articles, RCTs, and comparisons in the review. In the Abstract it is stated there were 8 RCTs with 9 comparisons. In the Results section of the main text, it states initially that there were 10 articles and 8 RCTs, although only 8 articles are cited. The following sentence then states there were 10 RCTs and 9 comparisons. However, the following sentences describe interventions for 8 studies, although there are 9 publications in Table 1.

We apologise for the confusion. There are nine interventions evaluated in 8 RCTs; these were reported in 10 articles. One RCT was reported in three separate papers and another RCT reported two eligible interventions. Unfortunately, in the Results section of the main text, paragraph 2, we stated in error:

After detailed evaluation, ten articles reporting data from eight RCTs⁵⁶⁻⁶³ were included in the review. These ten RCTs included data from 960 participants randomised to nine eligible comparisons. The interventions evaluated in the included studies were exercise, education, or a combination of both.

This should have read:

After detailed evaluation, ten articles reporting data from eight RCTs⁵⁶⁻⁶³ were included in the review. These eight RCTs included data from 960 participants randomised to nine eligible comparisons. The interventions evaluated in the included studies were exercise, education, or a combination of both.

R1: Abstract, line 41. The number of studies reported to evaluate exercise is incorrect.

We have changed “5 studies” to “6 interventions”

Inclusion of 1 further exercise intervention which could not be included in meta-analysis in the following sentence is clearly not helpful and is now reported after the meta-analysis results.

“Sensitivity analysis restricted to studies at overall low risk of bias confirmed findings. Another RCT of exercise with no data available for meta-analysis showed no benefit. Studies evaluating combined exercise and education intervention (n=1) and education alone (n=1) suggested similar findings.”

R1: Page 4, line 43-46. Stating that twelve of the reviews were older than 5 years and 5 only looked at short-term outcomes doesn't provide much useful information when we don't know what "numerous"

means. If 30 reviews had been conducted, this leaves a large number unaccounted for. Were these reviews searched for systematically, i.e. do the authors know they have an exhaustive list? This needs to be clarified to put these descriptions into context.

In our searches for RCTs we also looked for systematic reviews (lines 9 and 10 of MEDLINE strategy, provided in Appendix 1). The intention was primarily to check these reviews for RCTs that might not have been picked up in the search, as well as to help us build a context for this review. We have now briefly summarised all 36 relevant systematic reviews in paragraph 3 of the background:

“In a systematic literature search of MEDLINE, Embase, PsycINFO, CINAHL and The Cochrane Library on 18th December 2018, we identified 36 systematic reviews of pre-operative interventions in TKR. Twenty-five of these had searches conducted more than five years previously^{10,12-35}, and nine exclusively reported short-term outcomes (three months or less after surgery)³⁶⁻⁴⁴. Two reviews considered the outcome of long-term pain^{45,46} but neither had a registered protocol. Chesham and Shanmugam identified two randomised controlled trials in their search window (2004-2014) that

reported no difference in pain at six months after surgery in patients randomised to exercise or control⁴⁵. Wang and colleagues identified three randomised controlled trials of pre-operative exercise-based interventions in patients waiting for knee or hip replacement that reported long-term pain and were published up to November 2015⁴⁶. No difference in pain outcome was apparent at six months or longer in patients receiving intervention or control.”

R1: Page 4, lines 54-56. This sentence doesn't provide much useful information. What were the interventions? Does "small difference" mean significant but not clinically meaningful? Which groups had less pain? What does "not durable" mean?

We apologise for this being unclear. The review of Wang and colleagues is a well conducted systematic review and meta-analysis including studies to November 2015. However, no protocol was published, and they did not contact authors of RCTs for missing data. The authors included hip and knee replacement patients but did not report outcomes separately.

To make our interpretation of this study clearer, we have rewritten these sentences:

Wang and colleagues identified three randomised controlled trials of pre-operative exercise-based interventions in patients waiting for knee or hip replacement that reported long-term pain and were published up to November 2015²⁹. No difference in pain outcome was apparent at six months or longer in patients receiving intervention or control.

R1: Page 10, lines 11-13. There still appears to be some overlap in the intervention descriptions. What is the difference between range of movement, flexibility, and stretching exercises?

We reported the various components as described in individual studies. However, we have now grouped these under the specific aims of the intervention (additional to strengthening) as described in the articles:

56 D'Lima - main focus “strengthen the upper and lower extremities and improve knee range of motion”

57 Beaupre – main focus “to improve knee mobility or strength”

59 Huber – main focus “stability/postural function, functional alignment, lower extremity muscle strength and functional exercise”

60 Matassi – main focus muscle strength and flexibility 62
Rooks – main focus mobility

Text has been modified appropriately:

All exercise interventions included strengthening⁵⁶⁻⁶², with additional components targeting gait re-education⁵⁷, functional exercise^{57, 59}, improving range of movement, mobility or flexibility^{56, 57, 59, 60, 62}

and cardiovascular conditioning^{56, 59, 62}.

R1: Page 13, line 40. "either" implies two options.

Either is removed

R1: Page 14, lines 31-34. I am not sure what this sentence means.

To clarify this, we have modified the sentence to now read:

Previous reviews have focused on outcomes relating to pre-operative function and pain, the in-hospital experience, and long-term recovery^{12,13,15-29,69}. Review conclusions vary based on the outcomes and follow up times. In general, those concentrating on short term function and outcomes related to discharge are more likely to report early patient benefit and shorter hospital stay for patients receiving pre-surgical exercise and education compared with controls. Those considering longer-term post-operative outcomes show little difference in outcomes between randomised groups.

R1: Page 14, lines 47-49. Given that these two articles are the only differences from previous reviews, do they make any difference to the conclusions? It seems they continue to support previous research. The lower part of this paragraph discusses information that is more relevant to the strengths and weaknesses section. There is a lack of discussion on the findings of preoperative education - most of this section discusses previous reviews/studies on exercise.

To address these issues, we have added some details of how our review compares with previous reviews and moved the relevant part of the paragraph to the strengths and weaknesses section. This improves the structure of the discussion considerably and we are grateful for the suggestions and advice.

R1: Page 15, lines 34-39. I don't understand what the main point of this sentence is. Exercise and education were the only interventions included in the review.

This section is now removed.

The aim of our research, as described in our PROSPERO registration was “to assess the effectiveness of pre-operative interventions in preventing chronic pain after knee replacement.” The

interventions we sought were “Any therapeutic intervention (pharmacological or non-pharmacological) with potential for reducing chronic pain.” Those relating to implant design are being addressed in a separate project. Our review has some similarities with that of Wallis and Taylor 2011 which looked at a broad range of non-surgical and non-pharmacological pre-operative interventions. Our interest is on a similarly broad range of interventions but with a specific focus on the outcome of long-term pain. Mechanisms may be purely hypothetical, but patients and their treating healthcare professionals should be reassured that any treatment in the pre-, peri- and post- operative period is either of value in preventing long-term pain or safe with no associated adverse events such as long-term pain.

R1: Page 15, lines 41-48. I am not quite sure what the first sentence means. What is "lack of psychological support"? It should be noted that the study referred to here didn't actually manipulate pain catastrophizing, as there were no differences in this among the groups.

This is a fair comment. We have rewritten this and hope it summarises the issue of psychological distress before TKR and potential interventions.

An aspect of pre-surgical health with a possible link with chronic pain is psychological distress⁷¹. As an example, pain catastrophizing has been implicated as a risk factor for chronic pain⁷². However, a recent trial where intervention began in the pre-surgical period and extended well beyond it found that pain coping skills training for people with high pain catastrophizing did not improve WOMAC pain scores at 12 months after TKR⁷³. Other aspects of preoperative psychological distress are associated with long-term pain after TKR and evaluation of treatment strategies merits investigation⁷¹.

R1: Page 16, lines 6-10. Given that there was only 1 study that looked at education alone, would the authors not recommend further studies using education as an intervention? I am not sure what "in other areas" refers to in the following sentence.

We agree that the evidence base for education as a pre-operative intervention to help patients on the TKR pathway achieve good long-term outcomes is sparse. We have changed the conclusion appropriately. “In other areas” is removed.

Pre-surgical interventions have been evaluated in patients receiving TKR. We found low to moderate quality evidence to suggest that exercise is not effective in preventing long-term pain in adults scheduled to receive total knee replacement and it is difficult to recommend further research except of novel approaches and/or in specific populations. The evidence base for education and exercise combined with education was highly limited.

We thank Reviewer 1 for the thorough review of our research. We believe that the changes made have improved the article considerably.

VERSION 3 – REVIEW

REVIEWER	Gwyn lewis Auckland Univ Technol
REVIEW RETURNED	29-Nov-2019
GENERAL COMMENTS	The authors have nicely addressed my remaining comments. I commend them for a useful contribution to the literature.